# Understanding the uptake and determinants of prevention of mother-to-child transmission of HIV services in East Africa: Mixed methods systematic review and meta-analysis

Feleke Hailemichael Astawesegn[1,2,3]*, Haider Mannan[1], Virginia Stulz[4], Elizabeth Conroy[1]

1 Translational Health Research Institute (THRI), Western Sydney University, Campbelltown Campus, Penrith, New South Wales, Australia, 2 School of Public Health, College of Medicine and Health Sciences, Hawassa University, Hawassa, Ethiopia, 3 Rural Health Research Institute, Charles Sturt University, Orange, New South Wales, Australia, 4 School of Nursing and Midwifery Centre for Nursing and Midwifery Research, Western Sydney University, Kingswood, New South Wales, Australia

* felekeh86@gmail.com

## Abstract

### Background

Prevention of mother-to-child transmission (PMTCT) of HIV service is conceptualized as a series of cascades that begins with all pregnant women and ends with the detection of a final HIV status in HIV-exposed infants (HEIs). A low rate of cascade completion by mothers' results in an increased risk of HIV transmission to their infants. Therefore, this review aimed to understand the uptake and determinants of key PMTCT services cascades in East Africa.

### Methods

We searched CINAHL, EMBASE, MEDLINE, Scopus, and AIM databases using a predetermined search strategy to identify studies published from January 2012 through to March 2022 on the uptake and determinants of PMTCT of HIV services. The quality of the included studies was assessed using the Mixed Methods Appraisal Tool. A random-effects model was used to obtain pooled estimates of (i) maternal HIV testing (ii) maternal ART initiation, (iii) infant ARV prophylaxis and (iv) early infant diagnosis (EID). Factors from quantitative studies were reviewed using a coding template based on the domains of the Andersen model (i.e., environmental, predisposing, enabling and need factors) and qualitative studies were reviewed using a thematic synthesis approach.

### Results

The searches yielded 2231 articles and we systematically reduced to 52 included studies. Forty quantitative, eight qualitative, and four mixed methods papers were located containing evidence on the uptake and determinants of PMTCT services. The pooled proportions of maternal HIV test and ART uptake in East Africa were 82.6% (95% CI: 75.6–88.0%) and 88.3% (95% CI: 78.5–93.9%). Similarly, the pooled estimates of infant ARV prophylaxis and

**Data Availability Statement:** All relevant data are within the paper and its Supporting information files.

**Funding:** The author(s) received no specific funding for this work.

**Competing interests:** The authors have declared that no competing interests exist.

**Abbreviations:** AIDS, Acquired Immune Deficiency Syndrome; ANC, Antenatal care; AOR, Adjusted Odds Ratio; ART, Antiretroviral therapy; ARV, Antiretrovirals; CI, Confidence Interval; DBS, Dry blood spot; EID, Early Infant HIV Diagnosis; HCWs, Health Care Workers; HEIs, HIV Exposed Infants; HIV, Human Immunodeficiency Virus; MCH, Maternal and child health; MMAT, Mixed Methods Appraisal Tool; MTCT, Mother-To-Child Transmission of HIV; PITC, Provider Initiated HIV Testing and Counselling; PMTCT, Prevention of mother-to-child transmission of HIV; SES, socioeconomic status; SSA, Sub-Saharan Africa; UNAIDS, The United Nations Programme on HIV and AIDS.

EID uptake were 84.9% (95% CI: 80.7–88.3%) and 68.7% (95% CI: 57.6–78.0) respectively. Key factors identified were the place of residence, stigma, the age of women, the educational status of both parents, marital status, socioeconomic status, Knowledge about HIV/PMTCT, access to healthcare facilities, attitudes/perceived benefits towards PMTCT services, prior use of maternal and child health (MCH) services, and healthcare-related factors like resource scarcity and insufficient follow-up supervision.

## Conclusion

Most of the identified factors were modifiable and should be considered when formulating policies and planning interventions. Hence, promoting women's education and economic empowerment, strengthening staff supervision, improving access to and integration with MCH services, and actively involving the community to reduce stigma are suggested. Engaging community health workers and expert mothers can also help to share the workload of healthcare providers because of the human resource shortage.

## Background

The provision of PMTCT services plays a crucial role in the global fight against new HIV infections and to ensure an HIV/AIDS-free generation. For the last two decades, the implementation of the PMTCT program has substantially decreased the number of HIV-infected babies born to HIV-positive mothers worldwide. This progress has been achieved through the implementation of improved HIV diagnosis, care, and treatment services [1] with technical and financial support from international health organizations such as the world health organization (WHO) [2] and integration of maternal, newborn and child health services [3,4].

Despite considerable progress in the PMTCT program, HIV remains a disease of public health importance in sub-Saharan Africa (SSA) [5] where more than two-thirds of the world's HIV-infected children live [6]. The current global efforts in the fight against human immunodeficiency virus (HIV) have been focused on the virtual elimination of child HIV infection for resource-limited settings with targets of MTCT rate < 5% in breastfeeding countries and < 2% in non-breastfeeding countries [7–10]. Hence, World Health Organization endorsed lifelong antiretroviral therapy for pregnant and breastfeeding women diagnosed with HIV infection and provision of nevirapine to all HIV-exposed infants for 4–6 weeks (Option B+ approach) to prevent mother-to-child HIV transmission [11].

Effectively implemented PMTCT service can reduce the risk of vertical HIV transmission from 15–45% to less than 1% [12–14]. If PMTCT programs are going to be effective, HIV-infected pregnant women must be able to navigate through complex and sequential steps called the PMTCT cascade. It refers to the sequence of steps a mother with HIV takes from diagnosis through receiving appropriate care/treatment for themselves and their newborns [15–17]. It begins with all pregnant women and ends with exposed infants' HIV testing [17], and includes (but is not limited to) (1) maternal/prenatal HIV testing, (2) initiating antiretroviral therapy (ART) treatment for women identified as HIV positive as early as possible during pregnancy, birth, and breastfeeding, (3) ARV prophylaxis for HIV-exposed infants (HEI) within hours of birth; and (4) early infant HIV diagnosis (EID)/testing of infants at six weeks [18]. Thus, successful navigation of pregnant women along the PMTCT cascade is crucial and at each step, 95% uptake is required to effectively reduce MTCT of HIV and virtually eliminate MTCT by 2030 [5,11].

However, mother-to-child transmission (MTCT) of HIV has remained a challenge because of the cumulative dropout rate of women and their infants at each step along these cascades [19–21]. A review done in SSA showed that 94% of pregnant women were tested for HIV, 70% of those who were HIV-positive initiated ARV/ART, and 64% of the HEIs were tested for HIV at six weeks/ had an early diagnosis [22]. Health systems are challenged to support women's transfer along these various stages of care resulting in cumulative losses of pregnant women from the PMTCT program, with an increased risk of HIV transmission to their infants [23].

Studies worldwide have reported factors that have an association with maternal HIV test and ART services uptake, including place of residence [24–26], education level [24,26–28], maternal age [26,27], knowledge of HIV/AIDS and PMTCT [27,29,30], SES [24,26,31,32], lack of privacy and confidentiality [33], women's decision-making capacity [29,34,35], prenatal care [26,31], fear of disclosure and stigma [30,36–39]. Likewise, factors such as distance from a health facility [40], partner/family support [39], denial of HIV status [41], shortages of resources [41,42], lack of knowledge [43], feelings of guilt [43], children of known HIV positive fathers [44] and maternal receipt of ART/HAART [40,43,45] have also been reported to affect infant ARV prophylaxis and EID services uptake.

In East Africa, the uptake, and determinants of key PMTCT services cascades have not been collectively and systematically analysed and remain poorly understood. Therefore, this mixed-methods systematic review and meta-analysis aims to understand the uptake and determinants of (i) maternal HIV testing among pregnant/postpartum women, (ii) ART initiation among HIV-infected pregnant/postpartum women, (iii) initiation of ARV prophylaxis for HEIs, and (iv) EID/HIV test at six weeks of age/ in East Africa. With the current global aim of ending the HIV epidemic by 2030, providing such vital information would help policymakers design better strategies and implement targeted interventions in East Africa.

## Methods

This mixed-method systematic review and meta-analysis was conducted using the Preferred Reporting Items for Systematic Reviews and Meta-analyses (PRISMA) guidelines [46] [S1 Table].

### Inclusion criteria

This review considered quantitative, qualitative, and mixed methods studies. Studies with a quantitative study design (such as prospective cohort, retrospective cohort, case-control, or cross-sectional study) and a qualitative design (such as phenomenology, and grounded theory) were considered in the study. For the quantitative component of the review, the exposures of interest were factors that were associated with the key PMTCT cascade uptake. An exposure factor was identified when a study reported a statistically significant association between the exposure (independent) and the outcome (dependent) variable. In the qualitative component of the review, our interests were mothers' and providers' experiences and/or perceptions of the factors that affect PMTCT cascade uptake. Only studies conducted in East African countries (Burundi, Djibouti, Eritrea, Ethiopia, Kenya, Mauritius, Mayotte, Malawi, Mozambique, Réunion, Rwanda, Somalia, Sudan, South Sudan, United Republic of Tanzania, Uganda, Zambia, Zimbabwe, Madagascar, Seychelles, and Comoros) were taken into consideration [47,48]. Studies involving pregnant, postnatal, and breastfeeding mothers and/or providers of PMTCT services from East Africa were considered. Only full-text available studies, published in English, and published in a peer-reviewed journal from 01/01/2012 to 30/05/2022 were considered (this reflects the period after which option B+ strategy was introduced in East Africa by Malawi, the first country to do so [11,49]).

## Information sources and search strategies

Searches were performed in five databases PubMed, Scopus, EMBASE, African Index Medicus (AIM), and CINAHL. The search strategy used four search concepts including the following keywords: **Search #1**: "pregnant women", "HIV positive mother", "PMTCT mother", "Lactating mother", "Breast Feeding", "breastfeeding mother", "HIV exposed infant", "HIV exposed child". **Search #2**: "option b+", "b plus", "lifelong antiretroviral therapy", "universal antiretroviral therapy", PMTCT, "prevention mother-to-child transmission", "prevention mother to child transmission", "elimination of mother-to-child transmission", "elimination of mother to child transmission", "prevention of vertical Transmission", "prevention of parent to child transmission", "highly active Antiretroviral Therapy", HAART, "antiretroviral therap*", ART, "Triple Therapy", ARV, "antiretroviral", "anti-retroviral", "HIV test*", "opt-out HIV test*", "counselling and testing", VCT, "early infant diagnosis", "infant testing", "infant HIV testing". **Search #3**: uptake, utilization, factor*, correlates, determinant*, predicator*, facilitator*, barrier*. **Search #4**: "East Africa*", Burundi*, Djibouti*, Eritrea*, Ethiopia*, Kenya*, Mauritius*, Mayotte*, Malawi*, Mozambique*, Reunion*, Rwanda*, Somalia*, Sudan*, "South Sudan*", Tanzania*, Uganda*, Zambia*, Zimbabwe*, Madagascar*, Seychelles*, Comoros*. The four search concepts, their synonyms, and truncations by the use of the asterisk '*' where appropriate were combined using the Boolean operators 'OR', within concepts, and 'AND' to combine concepts to develop the final search strategy. The detailed search strategy can be found in S2 Table.

## Study selection and data extraction

All identified citations were collected and uploaded into the reference management software EndNote version X9, and duplicates were removed. Articles were further screened based on titles and abstracts followed by full-text assessment by two independent reviewers (F.H and T. Y). Any disagreements between the reviewers at each stage of the selection process were resolved through discussion. Using a structured table, the following information was extracted from each eligible article: (i) country; (ii) year of study; (iii) year of publication; (iv) study setting, (v) specific details about the participants (pregnant women, postpartum women, HEIs and PMTCT service providers); (vi) study methods/design (vii) sample size (viii) aim of the study and (ix) uptake and significant determinants or the phenomena of interest relevant to the review objective for qualitative studies.

## Assessment of methodological quality

The Mixed Methods Appraisal Tool (MMAT) version 2018 was used to evaluate the methodological quality of studies that met the inclusion criteria [50]. This tool has been utilized by various studies [51–53] and is designed to enable systematic reviewers to evaluate the methodological quality of different study designs (quantitative, qualitative, and mixed methods). The studies were categorized into three (high, moderate, and low) based on the number of criteria out of seven that were met (quality scores). Studies that scored 6 or more were considered high quality, studies that scored 3 to 5 were considered moderate quality, and studies that scored 2 or less were deemed low quality. Quality scores for each study are presented in S3 Table.

## Data analysis and synthesis

Firstly, we used the $I^2$ test statistic and its corresponding p-value to verify the heterogeneity among the studies that were included. A p-value of less than 0.05 was used as a threshold to determine whether or not heterogeneity was present [54]. We also used Egger's and Begg's

tests, as well as a funnel plot, to evaluate publication bias among the studies. Funnel plots are scatter plots that illustrate the effect size estimates (on the x-axis) versus the standard errors of the effect size (on the y-axis), with the estimated average effect size represented by a vertical line. Then, a random effects model was used to estimate the pooled proportion of HIV testing, ART initiation, infant prophylaxis, and EID uptake, due to the heterogeneity observed between studies (p < 0.01%). Additionally, for each cascade, subgroup analysis was performed by year of publication, study settings and countries, using the random effects model.

Secondly, The JBI method for conducting a mixed-method systematic review (MMSR) was used to examine the factors that influenced the uptake of PMTCT cascades [55]. This method involves conducting a separate analysis of quantitative and qualitative data, and then integrating the findings from both forms of evidence. Because of wide variations in the measurement of variables that affect PMTCT service uptake, it was not practical to conduct a meta-analysis to assess the effect of each factor. Hence, we performed narrative synthesis after tabulating individual studies based on their unique characteristics. The quantitative data synthesis was guided by Andersen's behavioural model i.e. community factors including health facility factors, predisposing factors, enabling factors and need factors including prior use services [56]. Therefore, significant factors across studies were matched to the appropriate category in the Andersen behavioural model. In this review, meta-analyses were carried out using R software version 4.2.1 to estimate PMTCT services uptake in East Africa.

## Result

### Study identification

A total of 2231 potentially relevant articles were retrieved from the literature search, with 1146 articles remaining after the deletion of duplicates. After title and abstract screening, 171 references were remained and included in the full-text screening. Finally, 52 studies were deemed eligible and included in this study (forty quantitative and eight qualitative and four mixed method articles) as illustrated with the PRISMA flow chart [Fig 1].

### Characteristics of included studies on prevention of mother-to-child transmission of HIV (PMTCT) services cascade

The studies published between January 2012 and March 2022 were carried out in a range of East African countries; namely, Burundi [57], Comoros [57], Ethiopia [25,45,57–74], Kenya [57,75–78], Malawi [25,45,57–74,79], Mozambique [57,76–78,80], Rwanda [57], South Sudan [81], Tanzania [82,83], Uganda [39,57,84–87], Zambia [57,88–90], and Zimbabwe [57,91–96]. Studies conducted among eligible pregnant/postpartum women, infants and service providers were included in the analysis. Almost all the studies (44/52) were facility-based [23,25,39,45,58–62,64–74,76–89,91–93,95–101]. Furthermore, the study design varied across studies: cross-sectional studies (n = 28); prospective cohort (n = 2); retrospective cohort (n = 6), retrospective chart review (n = 4), qualitative studies (n = 8) and mixed methods studies (n = 4). The mixed methods studies included a mix of surveys, interviews, and focus groups. The study characteristics are summarized in Table 1.

### Uptake of PMTCT of HIV services in East Africa

The pooled uptake for maternal HIV test, maternal ARV use, infant ARV prophylaxis and EID for the PMTCT of HIV in East Africa were estimated using a random effects model. Accordingly, the forest plots showed that the pooled uptake estimate was 82.69%; 95% CI: 75.62–88.03% for maternal HIV test [Fig 2]; 88.33% (95% CI: 78.59–93.98%) for maternal ART [Fig

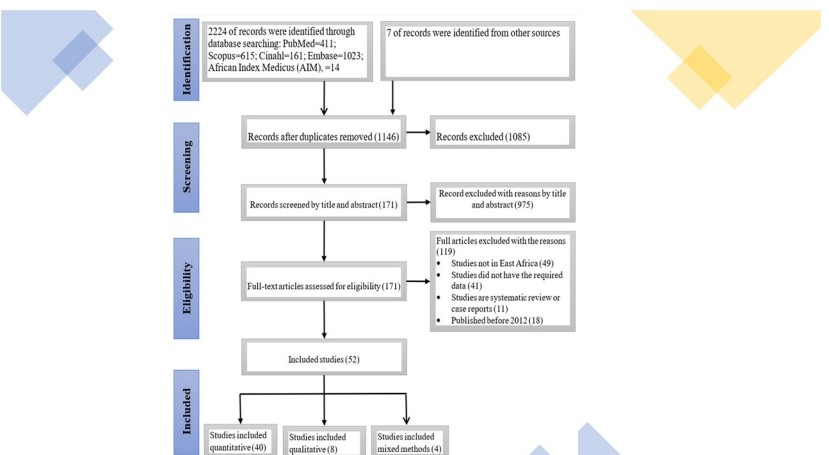

**Fig 1. PRISMA flow chart displaying studies that were identified, screened, and included in the review from databases.**

3]; 84.98% (95% CI: 80.78–88.39%) for infant ARV prophylaxis [Fig 4] and 68.77% (95% CI: 57.63–78.09%) for EID [Fig 5].

## Subgroup analysis

In this review, subgroup analysis was conducted for the country, study setting and year of publication. Accordingly, Malawi (98.8%) and Uganda (95.7%) had the highest proportion of women tested for HIV, while South Sudan (72%) and Ethiopia (75.5%) had the lowest proportion of women tested for HIV. The highest proportion of women who initiated ART was found in Mozambique (97.66%) followed by Malawi (95.76%), whereas the lowest proportion of women who initiated ART was found in Zambia (68.8%) followed by Zimbabwe (78.7%). Of the eight countries that had data on infant prophylaxis and EID uptake, Zimbabwe had the highest infant prophylaxis (91.9%) and EID uptake (85.6%).

Moreover, consistent maternal HIV testing was found across years, however, the highest (93.78%) and the lowest (73.7%) proportion of maternal ART uptake was found between the year 2012–2014 and 2015–2017 respectively. In all PMTCT cascades, studies that were conducted in the community had lower rates (77.9%, 95% CI: 69.7–84.3% for maternal HIV testing; 63.7%, 95% CI: 53.6–72.8% for maternal ART uptake, 62.8%, 95% CI: 60.1–65.3% for infant ARV prophylaxis) than studies that were conducted in the health facilities (84.7%, 95% CI:72.9–91.9% for maternal HIV testing, 91%, 95% CI:82.6–95.6% for maternal ART uptake, 86.8%,95% CI: 82.9–89.9% for infant ARV prophylaxis use) [S4 Table].

## Publication bias

To examine publication bias, we visually inspected the funnel plot shown in S1 Fig. A symmetric funnel plot indicates the absence of publication bias; that is, studies are evenly distributed on either side of the average effect size, regardless of the size of the study's sample. However, as shown in S1 Fig, the plots A to C are asymmetric. To test whether this asymmetry was statistically significant, we conducted Egger's regression test, which was significant for infant ARV prophylaxis uptake (z = 1.5725, p = 0.001) but not for maternal HIV testing, maternal ARV uptake and EID (z = 0.4437, p = 0.6572 for maternal HIV testing, z = 1.3556, p = 0.1752 for

**Table 1. Characteristics of included studies on prevention of mother-to-child transmission of HIV (PMTCT) services (n = 52).**

| First Author, publication year and reference | Aim(s) of the study/ Phenomena of interest | Country and study period | Study Population and sample size | Study setting | Study design | Targeted PMTCT cascade |
|---|---|---|---|---|---|---|
| Abtew 2015 [58] | To assess the acceptability of provider-imitated HIV testing and counselling (PITC) as an intervention for the PMTCT of HIV and to identify the associated factors | Ethiopia, 2014 | • Pregnant women attending antenatal care (ANC) services (n = 386) | Health facility | Cross-sectional | 1 |
| Ahoua 2020 [82] | To estimate the progress in the key indicators related to the PMTCT cascade in pregnant women enrolled in ANC services under Option B+ | Mozambique, 2013–2017 | • Pregnant women enrolled in ANC under Option B+ (n = 916280) | Health facility | Retrospective cohort | 1, 2 |
| Akal 2018 [59] | To determine the status of PMTCT services utilization and factors affecting PMTCT utilization in health facilities of Afar region, Ethiopia | Ethiopia, 2014–2015 | • Pregnant women attending ANC clinic (n = 347) <br> • Health professionals who were providing PMTCT services (n = 22) | Health facility | Cross-sectional | 1 |
| Alemu 2017 [25] | To identify the proportion of and factors for HIV testing among pregnant women in Ethiopia | Ethiopia, 2012–2013 | • Pregnant mothers (n = 416) | Health facility | Cross-sectional | 1 |
| Astawesegn 2021 [57] | To investigate the uptake and factors associated with prenatal HIV test uptake in East Africa. | Burundi 2016–2017, Comoros 2012, Ethiopia 2016, Kenya 2014, Malawi 2015–2016, Rwanda 2014–2015, Uganda 2016, Zambia 2013–2014, Zimbabwe 2015 | • Mothers who had birthed two years before the surveys- Burundi (n = 5412), Comoros (n = 1298), Ethiopia (n = 4308), Kenya (n = 7357), Malawi (n = 6693), Rwanda (n = 3236), Uganda (n = 5901), Zambia (n = 5074), Zimbabwe (n = 2454) | Community | Cross-sectional | 1 |
| Augustine 2021 [91] | To evaluate the cascade of HIV care among the HEIs born in Mashonaland East Province of Zimbabwe | Zimbabwe, 2017 | • HEIs (n = 1028) | Health facility | Retrospective review | 3,4 |
| Bergmann 2017 [84] | To identify influences on access to and use of infant HIV health services, specifically nevirapine administration. | Uganda, 2014–2015 | • HIV-positive pregnant women (n = 384) and six FGDs (n = 43) | Health facility | Mixed method | 3 |
| Berhan 2014 [60] | To assess the prevalence of HIV infection and associated factors among infants born to women living with HIV, in South Gondar zone, Amhara region, Ethiopia. | Ethiopia, 2013 | • HEIs (n = 434) | Health facility | Cross-sectional | 2,3,4 |
| Bobrow 2016 [97] | To understand barriers, facilitators, and recommendations for five key steps in the EID and treatment cascade: (1) identification of HEIs; (2) infant testing; (3) sample processing and transport; (4) reporting results to mothers; (5) ART initiation for HEI. | Malawi, 2013 | • Mothers of HEIs (n = 47) <br> • Healthcare workers (HCWs) providing EID and treatment (n = 20) | Health facility | Qualitative | 4 |
| Buleza Lamucene 2022 [80] | To understand the perspectives of pregnant and postpartum women living with HIV in Sofala, Mozambique, regarding barriers and facilitators to following PMTCT recommendations | Mozambique, 2020–2021 | • Pregnant and postpartum women living with HIV (n = 15) | Health facility | Qualitative | 1,2 |

(*Continued*)

**Table 1.** (Continued)

| First Author, publication year and reference | Aim(s) of the study/ Phenomena of interest | Country and study period | Study Population and sample size | Study setting | Study design | Targeted PMTCT cascade |
|---|---|---|---|---|---|---|
| Buregyeya 2017 [85] | To explore experiences of HIV-infected pregnant and breastfeeding women regarding barriers and facilitators to uptake and adherence to lifelong ART. | Uganda, 2014 | • HIV-infected pregnant and lactating women. (n = 57) | Health facility | Qualitative | 2 |
| Bwana 2018 [92] | To assess predictors of mothers/guardians to obtain EID services for children aged under 5 years exposed to HIV infection in Muheza district, Tanzania | Tanzania, 2015–2016 | • Mothers/guardians with children below 5 years who were born to HIV-positive mothers and were not breastfeeding for ≥6 weeks (n = 576). | Health facility | Cross-sectional | 4 |
| Cataldo 2017[98] | To explore the experience of patients and HCWs in relation to the implementation of Option B+. | Malawi, 2013. | • Nurses, medical assistants, and community health workers (Health Surveillance Assistants) (n = 48)<br>• HIV-infected pregnant or breastfeeding women (n = 24) | Health facility | Qualitative | 1,2 |
| Desta 2019 [61] | To determine the prevalence and associated risk factors of HIV among HEIs in the Tigray regional state, Northern Ethiopia | Ethiopia, 2016. | • HEIs (n = 340) | Health facility | Cross-sectional | 3 |
| Dzangare 2016 [99] | To assess the uptake and success of Option B+ in pregnant and breastfeeding women in two rural districts of Zimbabwe. | Zimbabwe, 2014 | • Women enrolled in ANC, labour and delivery care or post-natal care (n = 2598) | Health facility | Retrospective cohort | 1,2 |
| Ebuy 2020 [62] | To assess timely infant testing, testing for HIV at the 18th month, test results and factors influencing HIV positivity among infants born to HIV-positive mothers in public hospitals of Mekelle, Ethiopia. | Ethiopia, 2014–2017. | • Mother-infant pairs who were eligible for the PMTCT program (n = 558) | Health facility | Cross-sectional | 3,4 |
| Ejigu 2018[63] | To assess the uptake of HIV testing during pregnancy and associated factors among Ethiopian women. | Ethiopia, 2016 | • Women who were pregnant in the last year before the survey (n = 2414) | Community | Cross-sectional | 1 |
| Gaitho 2021[76] | To identify factors associated with timely uptake of virologic EID among HEI and gain insight into missed opportunities. | Kenya, 2015–2017 | • HEIs (n = 2020) | Health facility | Cross-sectional | 2,3,4 |
| Gamell 2017 [93] | To describe the PMTCT cascade and uptake of Option B+ guidelines implemented through this service delivery model | Tanzania, 2014–2015 | • Pregnant women (n = 1,579), HEIs (n = 135) | Health facility | Prospective cohort | 1,3 |
| Gebeyehu 2019[64] | To assess the acceptance of HIV testing and associated factors among pregnant women | Ethiopia, 2019 | • Pregnant women attending ANC (n = 340) | Health facility | Cross-sectional | 1 |
| Gebremedhin 2018[65] | To describe the level of acceptance of PITC and associated factors among pregnant women attending 8 ANC clinics in Adama, Ethiopia. | Ethiopia, 2016 | • Pregnant women attending ANC (n = 441) | Health facility | Cross-sectional | 1 |
| Gebresillassie 2019[66] | To assess the utilization and acceptance rate of PICT as an intervention for PMTCT among pregnant women attending University of Gondar referral and teaching hospital, Ethiopia. | Ethiopia, 2018 | • Pregnant women attending ANC (n = 364) | Health facility | Cross-sectional | 1 |

(*Continued*)

**Table 1.** (*Continued*)

| First Author, publication year and reference | Aim(s) of the study/ Phenomena of interest | Country and study period | Study Population and sample size | Study setting | Study design | Targeted PMTCT cascade |
|---|---|---|---|---|---|---|
| Haider 2022 [75] | To examine the factors affecting HIV testing among women during pregnancy in Kenya. | Kenya, 2014 | • Pregnant women (n = 36,626) | Community | Cross-sectional | 1 |
| Hampanda 2017[89] | To explore how gender power dynamics within couples affect HIV-positive women's uptake of early infant HIV testing at a large health centre in Lusaka, Zambia. | Zambia, 2014 | • HIV-positive mothers who had brought their child for routine pediatric immunizations (n = 320) | Health facility | Cross-sectional | 4 |
| Kanguya 2022 [88] | To understand readiness to start ART among HIV pregnant women from the perspectives of both women and men to suggest more holistic programs to support women to continue life-long ART after delivery. | Zambia, 2015 | • HIV-infected pregnant women not yet on ART and HIV-infected pregnant or postnatal women on ART (n = 20), Partners of women who were recently or currently pregnant (n = 16) | Health facility | Qualitative | 2 |
| Kebede 2014[45] | To investigate the rate of EID and predictive factors of EID among infants born to HIV-infected women | Ethiopia, 2012 | • Mother-infant pairs (n = 266) | Health facility | Retrospective cohort | 3,4 |
| Konje 2018 [94] | To conduct a population-based study that examined the utilization and availability of ANC services | Tanzania, 2016–2017 | • Pregnant women (n = 1719) | Community | Mixed method | 1 |
| Lain 2020 [83] | To describe the completeness of follow-up until definitive diagnosis among HEI, who were enrolled in routine care, the presence of clinical events during follow-up and to analyse factors associated with LTFU and clinical events. | Mozambique, 2019 | • HEIs (n = 1413) | Health facility | Retrospective cohort | 2,3 |
| Makau 2015 [77] | To evaluate determinants of EID and early treatment initiation among HIV-exposed children from informal settlements in Nairobi, Kenya. | Kenya, 2013 | • HIV-infected mother-infant pairs (n = 238) | Health facility | Cross-sectional | 3,4 |
| Moges 2017 [67] | To determine the rate of HIV transmission and associated factors among HEIs in selected health facilities in East and West Gojjam Zones, Northwest Ethiopia | Ethiopia, 2015 | • HEI-mother pairs (n = 305) | Health facility | Retrospective cohort | 3,4 |
| Mukose 2021 [86] | To assess key issues around ART prescription and swallowing (uptake), early adherence and associated factors among HIV-positive expectant women and lactating mothers on Option B+ in Central Uganda. | Uganda, 2013–2016. | • HIV-positive pregnant women (n = 507), HIV-positive pregnant and breastfeeding women (n = 57), Health provider(n = 54) | Health facility | Mixed methods | 2 |
| Mustapha 2018 [39] | To evaluate the utilization of PMTCT services and associated factors among adolescent and young postpartum mothers aged 15 to 24 years at a public urban referral hospital in Uganda. | Uganda, 2015 | • Postpartum mothers (HIV positive and negative) (n = 418), <br>• For In-depth interviews (10 HIV positive and 10 HIV negative) postpartum mothers, for key informants (2 nurses, 2 counsellors, 3 peer educators, and 2 doctors) | Health facility | Mixed methods | 1,2 |

(*Continued*)

**Table 1.** (Continued)

| First Author, publication year and reference | Aim(s) of the study/ Phenomena of interest | Country and study period | Study Population and sample size | Study setting | Study design | Targeted PMTCT cascade |
|---|---|---|---|---|---|---|
| Ng'ambi 2022 [79] | To describe HIV prevalence trends and assess the factors associated with the risk of HIV infection of HEI tested with DNA-PCR in Malawi between 2013 and 2020. | Malawi, 2013–2020 | • HEIs (n = 255 229) | Health facility | Retrospective review | 4 |
| Nungu 2019 [95] | To determine the re-testing uptake and the determinants of HIV re-testing in a rural region of Njombe in Tanzania. | Tanzania, 2015–2016 | • Newly delivered mothers (≤7 days from delivery) (n = 668) | Health facility | Cross-sectional | 1 |
| Ongaki 2019 [78] | To determine factors affecting uptake of PMTCT services among HIV-positive pregnant women at Lodwar County Referral Hospital in Turkana County, an arid area in northern Kenya. | Kenya, 2015–2016 | • HIV-positive pregnant mother (n = 230) | Health facility | Retrospective review | 2,3 |
| Oshosen 2021[96] | To elicit the perspectives of PMTCT patients regarding the content and quality of the counselling they received during HTC in Tanzania. | Tanzania, 2016–2017. | • HIV-positive pregnant women (n = 24) | Health facility | Qualitative | 1,2 |
| Tadewos 2020[68] | To assess the proportion of mother-to-child transmission (MTCT) of HIV and associated factors among HEIs on follow-up in pastoralist health facilities, South Omo, Ethiopia. | Ethiopia, 2018 | • HEI-mother pairs who were on follow-up care (n = 228) | Health facility | Cross-sectional | 3,4 |
| Thidor 2019[81] | To assess knowledge, attitude and practice of prevention of MTCT of HIV among pregnant women attending ANC at Juba Teaching Hospital, South Sudan | South Sudan, 2015 | • Pregnant women (n = 251) | Health facility | Cross-sectional | 1 |
| Tsehay 2019 [69] | To assess factors associated with HIV-positive serostatus among HEIs attending care at health facilities in Bahir Dar administration, Ethiopia | Ethiopia, 2018 | • HEIs (n = 477) | Health facility | Cross-sectional | 3,4 |
| Van Lettow 2018 [23] | To estimate the use and outcomes of the Malawian programme for the MTCT of human immunodeficiency virus (HIV) | Malawi, 2014–2016 | • Mother-infant pairs (n = 33744) | Health facility | Cross-sectional | 1,2,3,4 |
| Wanyenze 2018 [87] | To assess the uptake of PMTCT services in a cohort of HIV-infected women in care at The AIDS Support Organization Jinja and Kampala in Uganda. | Uganda, 2013 | • HIV-infected women with fertility intentions (n = 299) | Health facility | Prospective cohort | 2 |
| Workagegn 2015 [70] | To identify predictors and possible barriers to HIV testing among ANC attendees based on the health belief model (HBM) in Addis Ababa, Ethiopia. | Ethiopia, 2013 | • Pregnant women attending ANC (n = 301) | Health facility | Cross-sectional | 1 |
| Wudineh 2016 [71] | To investigate mother-to-child transmission (MTCT) of HIV infection and its determinants among HEIs on care at Dilchora Referral Hospital in Dire Dawa City Administration. | Ethiopia, 2013 | • HEIs (n = 382) | Health facility | Retrospective cohort | 2,3,4 |

(*Continued*)

**Table 1.** (Continued)

| First Author, publication year and reference | Aim(s) of the study/ Phenomena of interest | Country and study period | Study Population and sample size | Study setting | Study design | Targeted PMTCT cascade |
|---|---|---|---|---|---|---|
| Zegeye 2020 [72] | To assess the PMTCT service utilization rate and to characterize its reasons among pregnant women attending ANC clinics at selected public health facilities in Debre Berhan Town, Northern Ethiopia. | Ethiopia, 2019 | • Pregnant women attending ANC (n = 355) | Health facility | Cross-sectional | 1 |
| Chadambuka 2018 [100] | To explore the acceptability of Option B+ among pregnant and lactating women in Zimbabwe | Zimbabwe, 2014–2015 | • HIV-positive pregnant and breastfeeding women. (n = 43 for In-depth interview, n = 22 for Focus group interview) | Health facility | Qualitative | 2 |
| Semali 2014 [102] | To determine factors associated with uptake of HIV testing during ANC in Tanzania. | Tanzania, 2011–2012 | • Women who attended antenatal clinic (ANC) and gave birth in the past two years (n = 3555) | Community | Cross-sectional | 1 |
| Deressa 2014 [74] | To investigate factors associated with the acceptability and utilization of PMTCT of HIV. | Ethiopia, 2010 | • Pregnant women attending ANC (n = 843) | Health facility | Cross-sectional | 1 |
| Ford 2018 [90] | To investigate associations between decision-making and specific steps along the PMTCT cascade. | Zambia, 2011 | • HIV-infected mother-infant pairs (n = 344) | Community | Cross-sectional | 2,3 |
| Olana 2016 [73] | To assess the proportion of HIV-infected babies tested by DNA-PCR and factors affecting HIV transmission | Ethiopia, 2014 | • HEIs (n = 624) and their mothers (n = 412) | Health facility | Retrospective review | 4 |
| Yaya 2019 [103] | To assess the sociodemographic and economic factors associated with ANC use and HIV testing during pregnancy in Mozambique. | Mozambique, 2011 | • Women who were pregnant in the last two years before the survey (N = 7080) | Community | Cross-sectional | 1 |
| McCoy 2015 [104] | To examine the uptake of services and behaviours in the prevention of mother-to-child HIV transmission (PMTCT) cascade in Zimbabwe and determine factors associated with MTCT and maternal ART or ARV prophylaxis. | Zimbabwe, 2012 | • Mothers/caregivers-infant's pair (n = 8,800) | Community | Cross-sectional | 1,2,3 |
| Kim 2016 [101] | To identify the main barriers and facilitators to uptake and adherence to ART under Option B+ | Malawi, 2014 | • HIV-positive pregnant and postpartum women (n = 65) | Health facility | Qualitative | 2 |

Targeted PMTCT cascade: 1 = Maternal HIV testing; 2 = Maternal antiretroviral therapy; 3 = Infant antiretroviral prophylaxis, 4 = Early infant HIV diagnosis.

maternal ARV uptake, z = -0.5671, p = 0.5706 for EID). Our further sensitivity analysis (i.e., trim-and-fill analysis) for infant prophylaxis did not reveal a significant influence on pooled uptake.

## Reported factors associated with PMTCT of HIV services uptake in East Africa

**Quantitative synthesis.** Table 2 shows the factors that the quantitative studies have found to be statistically significantly associated with maternal HIV test, maternal ARV, infant ARV prophylaxis, and EID uptake. Factors from quantitative studies were organized into four

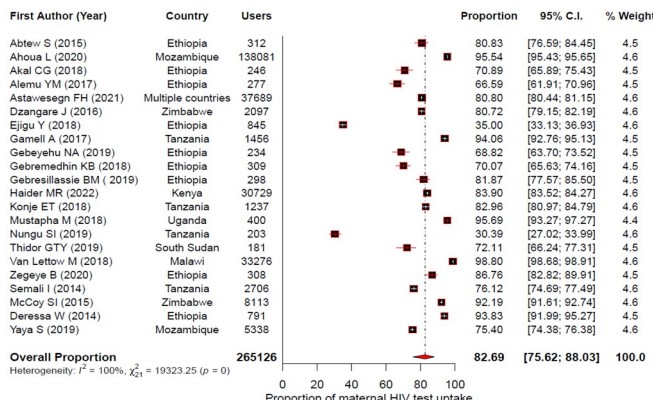

**Fig 2. Forest plot showing individual studies and pooled estimates of maternal HIV test uptake in East Africa based on a random-effects model.**

categories according to Andersen's behavioural model: community and health care factor, predisposing factors, enabling factors, need and prior health service use factors [Table 2].

**Community and health care factors**: Six studies [25,57,58,63,66,103] showed the presence of an association between place residence and maternal HIV testing. However, two studies on maternal HIV testing [65,75] and three studies on EID [62,91,92] revealed the absence of statistical association with place of residence. Women who lived in a community where there was no/low stigmatized attitude toward people living with HIV/AIDS [25,58] were more likely to be HIV tested than those who lived in a community that stigmatized HIV-positive patients [63,75]. Two studies explicitly indicated an association between the quality of available HIV testing services and maternal HIV test uptake [65,95].

**Predisposing factors**: Eight studies reported the presence of an association [25,57,59,63,64,66,95,103] and three studies [65,70,75] reported the absence of an association between educational level and maternal HIV testing. Of the eight studies that showed the presence of association, seven studies reported more educated women were found to be HIV tested than non-educated women [25,57,59,63,64,95,103]. Similarly, for women who have educated partners [57,103] and exposure to media [57,103] the odds of HIV testing were higher. Furthermore, a study conducted in Tanzania showed the presence of a correlation between children's age and marital status with EID [92].

**Enabling factors**: Factors under the enabling category included socioeconomic factors (for example, employment, higher monthly income and wealth index) [25,57–59,63,66,95,103],

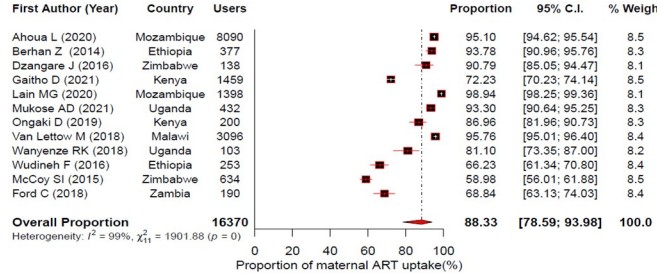

**Fig 3. Forest plot showing individual studies and pooled estimates of maternal ART uptake in East Africa based on a random-effects model.**

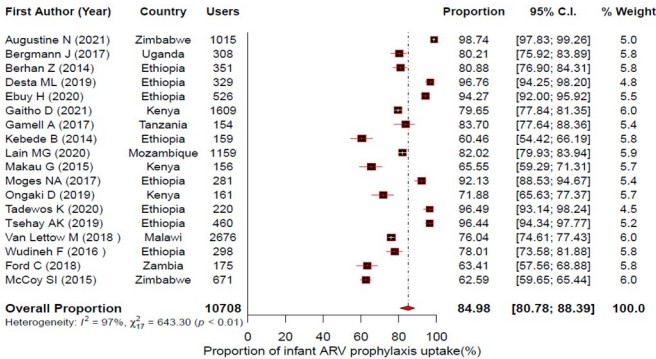

**Fig 4. Forest plot showing individual studies and pooled estimates of infant ARV prophylaxis uptake in East Africa based on a random-effects model.**

access to the PMTCT services [57,59,92], knowledge/awareness about HIV/PMTCT/MTCT [25,57,63–65,75,92], and presence of support groups [84]. Studies showed that higher SES was positively associated with HIV testing. For example, having a higher wealth or monthly income was positively related to taking HIV testing multiple times [57,63,103]. Moreover, occupation or employment was found to be correlated with maternal HIV testing in four studies [58,66,95,103]. Nevertheless, three studies indicated that there was no significant association between occupation or employment and maternal HIV testing [57,63,64]. Similarly, comprehensive knowledge about PMTCT is significantly associated with a higher rate of HIV testing [25,57,63,64,75]. Only one Ugandan study reported associations between social support and infant ARV prophylaxis uptake [84] and infants from mothers who are currently in a social support group were more likely to be HIV tested than their counterparts (AOR = 2.50) [84]. Nevertheless, women who have lower access to PMTCT services were less likely to use both maternal HIV testing and EID [57,59,92].

**Need factors and prior health services use**: Women who did not want to have more children were less likely to utilize maternal HIV testing [103]. However, women who had symptoms of sexually transmitted infections [95] and more ANC visits [64,65,95] were found to be more likely to use the HIV test service. Women who were on ART during pregnancy or at the time of the HIV PCR test and infants who had ARV prophylaxis at birth were positively associated with EID [45,76]. Likewise, factors such as ANC follow-up [62], birth at a government health facility [45], and maternal ART adherence [89] were found to be positively correlated

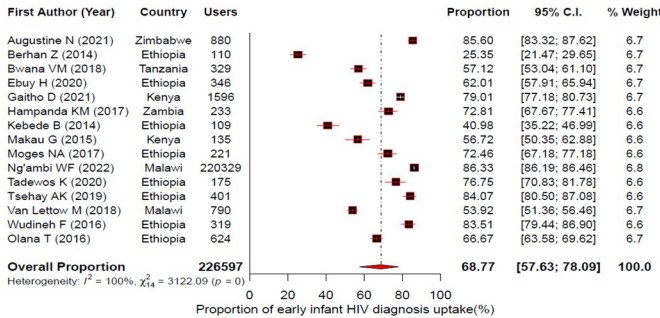

**Fig 5. Forest plot showing individual studies and pooled estimates of early infant HIV diagnosis uptake in East Africa based on a random-effects model.**

**Table 2.** Statistically significant factors affecting prevention of mother-to-child transmission of HIV services uptake from multivariable analyses in East Africa.

| PMTCT Cascades | First Author, publication year and reference | Factors affecting prevention of mother-to-child transmission of HIV services uptake | | | | |
|---|---|---|---|---|---|---|
| | | Community and Health care factors* | Predisposing factors* | Enabling factors* | Need factors and prior use of services* | Controlled variables during multivariate analysis |
| Maternal HIV testing | Abtew 2015 [58] | • Residents: Rural (AOR = 4.04; 95% CI: 1.24–13.11), Urban (ref) <br>• Stigma toward people living with HIV/AIDS: No (AOR = 3.54; 95% CI: 1.23–10.16), Low (AOR = 4.04; 95% CI: 1.52–10.72), High (ref) | • Attitudes towards PITC: Favourable (AOR = 1.57; 95% CI: 1.08–6.25), Not favourable (ref) <br>• Perceived pre-test counselling service: Good (AOR = 4.23; 95% CI: 2.01–8.89), Poor (ref) | • Occupation: employed (AOR = 2.15; 95% CI:1.08–4.30), Students (AOR = 6.00; 95% CI:1.45–24.75), merchants (AOR = 4.43; 95% CI:1.18–16.68), housewives <br>• Planned to disclose their test results to their husbands: Yes (AOR = 14.85;95% CI:4.60–47.94), No (ref) | | Partner reaction to the positive result, access to health facilities for HIV testing, access to transport services, preferred sex, and age of counsellor. |
| | Alemu 2017 [25] | • Residents: Urban (AOR = 3.42; 95% CI: 1.82–6.46), Rural (ref) <br>• Favoured attitude toward persons living with HIV: Yes (AOR = 2.42; 95% CI: 1.20–4.86), No (ref) | • Women aged: 16–24 years (AOR = 7.9; 95% CI: 3.19–19.55), 25–29 years (AOR = 4.95; 95% CI: 2.06–11.88), 30–34 years (AOR = 3.31; 95% CI: 1.27–8.60), ≥ 35 years (ref) <br>• Maternal education: Secondary education and above (AOR = 3.49; 95% CI: 1.56–7.77), No formal education (ref) <br>• Parity: No child (AOR = 4.34; 95% CI: 1.61–11.68), 1–4 (AOR = 4.7; 95% CI: 1.94–11.36), ≥5 | • Monthly expenditure: ≥30 USD (AOR = 4.06; 95% CI: 1.66–9.93), < 30 USD (ref) <br>• Comprehensive knowledge about MTCT: Yes (AOR = 3.73; 95% CI: 1.56–8.94), No (ref) <br>• Comprehensive knowledge about PMTCT (AOR = 2.56; 95% CI: 1.26–5.19), No (ref) | | No controlled variables |
| | Astawesegn 2021[57] | • Resident: Rural (AOR = 0.66; 95% CI: 0.51–0.85), Urban (ref) <br>• Countries: Comoros (AOR = 0.007; 95%CI:0.005–0.01), Ethiopia (AOR = 0.04; 95% CI: 0.03–0.05), Mozambique (AOR = 0.15, 95%CI: 0.11–0.20), Rwanda (AOR = 2.30; 95%CI: 1.29–4.08), Uganda (AOR = 0.61; 95%CI: 0.46–0.82), Zambia (AOR = 0.54; 95%CI:0.39–0.73), Zimbabwe (AOR = 0.42; 95% CI: 0.28–0.64), Burundi (ref) | • Maternal education: primary education (AOR = 1.29; 95% CI:1.10–1.50), secondary or higher education (AOR = 1.96; 95% CI:1.53–2.51), No education (ref) <br>• Partners education: primary education (AOR = 1.24; 95% CI:1.06–1.45), secondary or higher school (AOR = 1.56; 95% CI:1.26–1.94), No education (ref) <br>• Read a magazine/newspaper: Yes (AOR = 1.31; 95% CI:1.04–1.65), No (ref) <br>• Watch television: Yes (AOR = 1.46; 95% CI:1.20–1.79), No (ref) <br>• Listen to the radio: Yes (AOR = 1.13; 95% CI:1.01–1.29), No (ref) | • Distance to health facility: Challenging (AOR = 0.80; 95% CI:0.69–0.91), Not challenging (ref) <br>• Wealth index: rich (AOR = 1.57; 95% CI:1.17–2.11); middle AOR = 1.29; 95% CI:1.11–1.50), Poor (ref) <br>• Awareness about MTCT of HIV during birth: Yes (AOR = 1.73; 95% CI:1.42–2.10), No (ref) <br>• Awareness about MTCT of HIV during breastfeeding: Yes (AOR = 1.41; 95% CI:1.16–1.71), No (ref) | | Maternal age, Maternal occupation, History of Sexual violence, Household Decision making, Desire for pregnancy |
| | Ejigu 2018 [63] | • Residence: Urban (AOR = 3.30; 95% CI: 1.39–7.85), Rural (ref) <br>• Region: Tigray (AOR = 9.55; 95% CI: 5.17–17.64), Amhara (AOR = 4.16; 95% CI: 2.22–7.79), Addis Ababa/Harari/Dire Dawa (AOR = 3.55; 95% CI: 1.83–6.90), Afar/ Somali (ref) <br>• Stigmatizing attitude towards HIV-positive people: Yes (AOR = 0.57; 95% CI: 0.40–0.79)), No (ref) | • Maternal education: Primary level (AOR = 1.55; 95% CI: 1.12–2.15), Secondary level (AOR = 2.56; 95% CI: 1.36–4.82), Higher level (AOR = 3.95; 95% CI: 1.31–11.95), No education (ref) <br>• Marital status: Widowed/ Divorced/Separated (AOR = 0.07; 95%CI: 0.01–0.46), Never married (ref) | • Women who were aware of MTCT during pregnancy (AOR = 2.03; 95%CI: 1.48–2.78) <br>• Wealth index: Poorer (AOR = 2.32; 95% CI: 1.47–3.66), Middle (AOR = 3.27; 95% CI: 1.94–5.52), Richer (AOR = 5.43; 95% CI: 3.31–8.89), Richest (AOR = 5.84; 95% CI: 2.99–11.43), Poorest (ref) | | Maternal age, religion, employment status |
| | Gebeyehu 2019 [64] | | • Educational status: No formal education (AOR = 0.39; 95% CI: 0.16–0.96), Primary school (AOR = 0 35; 95% CI: 0.15–0.84), Tertiary school and above (ref) <br>• Perceived the benefit of HIV testing: Yes (AOR = 1.83; 95% CI: 1.08–3.10), No (ref) | • Awareness about MTCT: Yes (AOR = 3.77; 95% CI: 2.12–6.72), No (ref) <br>• Knowledge of PMCT: Knowledgeable (AOR = 1.71; 95% CI: 1.03–2.85), not Knowledgeable (ref) | • Number of ANC visits: Two or more ANC visits (AOR = 2.48, 95% CI: 1.46–4.22), One visit (ref) | Occupation, heard from a friend, heard from a health provider |
| | Akal 2018 [59] | | • Educational status: Could read and write (AOR = 11.3; 95% CI: 2.8–15, Could not read and write (ref) | • Income level per month: ≥1000 ETB (AOR = 5.6; 95% CI: 2.05–15.03), <1000 ETB (ref) <br>• Access to the PMTCT service: Yes (AOR = 6.5; 95% CI: 2.35–18.4), No (ref) | | Knowledge about PMTCT service |

(Continued)

**Table 2.** (Continued)

| PMTCT Cascades | First Author, publication year and reference | Factors affecting prevention of mother-to-child transmission of HIV services uptake | | | | |
|---|---|---|---|---|---|---|
| | | Community and Health care factors* | Predisposing factors* | Enabling factors* | Need factors and prior use of services* | Controlled variables during multivariate analysis |
| | Gebremedhin 2018 [65] | • Quality of PITC service (AOR = 1.91; 95% CI: 1.19–3.08) | • Older age (AOR = 0.37; 95% CI: 0.19–0.74 | • Higher level of knowledge on MTCT (AOR = 1.82; 95% CI: 1.03–3.20) <br> • Partner attitude toward positive HIV result: negative (AOR = 0.31; 95% CI: 0.10–0.94), positive (ref) | • More ANC visits (AOR = 2.59; 95% CI: 1.01–6.63) | Residence, educational status, monthly income, availability of HIV testing and counselling service, gender preference of the counsellor, stigmatizing attitude |
| | Gebresillassie 2019 [66] | • Resident: Rural (AOR = 3.64; 95%CI: 2.17–6.34), Urban (ref) | • Educational status: No formal education (AOR = 3.15; 95% CI:1.86–6.82), Primary education status (AOR = 2.73; 95% CI:1.17–5.43), Tertiary education (ref) | • Higher average monthly income (AOR = 4.01; 95%CI: 2.32–7.61), <br> • Employment status: self-employed (AOR = 0.31; 95% CI: 0.12–0.85), Unemployed (ref) <br> • Planned to disclosure HIV test results to male partners: Yes (AOR = 7.81; 95%CI:3.17–13.14), No (ref) | | Client age in year, religion, parity, partner reaction to a positive result, attitude toward PICT |
| | Haider et al 2022 [75] | • HIV-related stigma (AOR = 0.83; 95% CI:0.73–0.96), <br> • Regions: North-eastern (AOR = 0.33; 95% CI: 0.16–0.68), Eastern (AOR = 0.53; 95% CI: 0.30–0.95), Central (AOR = 0.45; 95% CI: 0.22–0.92), Western (AOR = 0.42; 95% CI: 0.22–0.81), Coast (ref) <br> • HIV counselling during ANC visits: Yes (AOR = 1.89; 95% CI: 1.39–2.56), No (ref) | • Maternal age: 20–29 years (AOR = 0.51; 95% CI: 0.29–0.91), 30–39 years (AOR = 0.31; 95% CI: 0.17–0.57), 40–49 years (AOR = 0.35; 95% CI: 0.16–0.75), 15–19 years (ref) <br> • Marital Status: Married and living with a partner (AOR = 1.48; 95% CI: 1.06–2.06), Never in union/ Widowed/Separated/ Divorced) (ref) | • Knowledge about HIV Transmission during pregnancy (AOR = 1.19; 95% CI: 1.05–1.34) | | Maternal education, Religion, Residence, and Wealth Index |
| | Nungu 2019 [95] | • Quality of HIV testing: Yes (AOR = 2.1; 95%CI:1.53–3.04), No (ref) | • Education: High Education (AOR = 1.9; 95%CI:1.25–3.02), Low Education (ref) | • Employment status: employed (AOR = 2.1; 95%CI: 1.06–4.34), unemployed (ref) | • Reported symptoms of sexually transmitted infections: Yes (AOR = 4.9; 95%CI: 2.15–6.14), No (ref) <br> • Number of ANC visit: ≥4 visits (AOR = 1.8; 95%CI: 1.21–2.69), < 4 (ref) <br> • Ever used condom: Yes (AOR = 1.7; 95%CI: 1.13–2.71), No (ref) | Parity, perceived severity of HIV |
| | Workagegn 2015 [70] | | • Maternal age: 21–25 years (AOR = 2.43; 95% CI: 1.13–5.23), 26–30 years (AOR = 2.3; 95%CI: 1.08–4.88), > 35 years (ref) <br> • Perceived net benefits: low (AOR = 0.34; 95%CI:0.19–0.58), high (ref) <br> • Perceived self-efficacy: high (AOR = 1.90; 95%CI: 1.09–3.33), low(ref) | | | Educational status, marital status, perceived threat, cues to action |
| | Yaya 2019 [103] | • Resident: Rural (AOR = 0.78; 95% CI: 0.68–0.90), Urban (ref) <br> • Ethnicity: Portugais (AOR = 2.09; 95% CI: 1.73–2.53), Xichangana (AOR = 2.07; 95% CI: 1.71–2.50), Cisena (AOR = 1.38; 95% CI: 1.20–1.58), Elomwe (AOR = 0.41; 95% CI: 0.31–0.55), Cindau (AOR = 1.41; 95% CI: 1.09–1.82), Xitswa (AOR = 6.63; 95% CI:4.48–9.81), Other (AOR = 3.16; 95% CI: 2.26–4.43), Emakhuwa (ref) <br> • Religion: Other (AOR = 0.86; 95% CI: 0.76–0.98), Islam (ref) | • Maternal age (AOR = 1.23; 95% CI:1.03–1.46 for age 20–24, AOR = 1.42; 95% CI: 1.19–1.69 for age 25–29, AOR = 1.37; 95% CI: 1.13–1.66 for age 30–34, for age 35–39, AOR = 1.28; 95% CI: 1.03–1.58, age 15–19 (ref)), <br> • Maternal education: Primary (AOR = 1.37; 95% CI:1.22–1.53), Secondary (AOR = 1.39; 95% CI: 1.17–1.65), No education(ref) <br> • Husband education: Primary (AOR = 1.22; 95% CI:1.08–1.37), Secondary (AOR = 1.45; 95% CI: 1.25–1.68), No education(ref) <br> • Media access: yes (AOR = 1.23; 95% CI: 1.10–1.38), no (ref) | • Employment: agricultural (AOR = 0.74; 95% CI: 0.66–0.82), professional/technical/managerial employed (AOR = 0.86; 95% CI: 0.74–0.99), Not working (ref) <br> • Wealth quantile: middle (AOR = 1.24; 95% CI:1.08–1.42), richer (AOR = 1.42; 95% CI: 1.21–1.67), richest (AOR = 1.95; 95% CI: 1.57–2.43), poorest (ref) | • Wanted more child: No (AOR = 0.64; 95% CI:0.44–0.92), Yes (ref) | Parity, household head's sex |

*(Continued)*

**Table 2.** (Continued)

| PMTCT Cascades | First Author, publication year and reference | Factors affecting prevention of mother-to-child transmission of HIV services uptake | | | | |
| --- | --- | --- | --- | --- | --- | --- |
| | | Community and Health care factors* | Predisposing factors* | Enabling factors* | Need factors and prior use of services* | Controlled variables during multivariate analysis |
| Infant ARV prophylaxis | Bergmann 2017 [84] | | | • Involving in the support group (AOR = 2.50; 95%CI: 1.06–5.83) | • Delivered at home (AOR = 0.02; 95% CI: 0.003–0.09)<br>• Mothers who took < 95% of ART doses in the last 30 days (AOR = 3.55; 95% CI: 1.36–9.26), | Maternal age, marital status, education, number of children, disclosure of HIV status to partner, and health care facility, |
| Early infant HIV diagnosis | Augustine 2021 [91] | | | | • Mode of delivery: non-normal vertex delivery (RR = 1.19; 95%CI: 1.01–1.40), normal vertex delivery (ref) | Sex, Place of birth, Birth weight (g), Place of residence, Type of health facility, District |
| | Bwana 2018 [92] | | • Married/living together with their spouses (AOR = 2.3; 95%CI: 1.2–4.6), Separated/divorced/widow (ref)<br>• Child age in months: 13–59 months (AOR = 0.4; 95%CI: 0.2–0.7), ≤12 months (ref) | • Living far away from the health facility (AOR = 0.6; 95%CI: 0.4–0.9)<br>• Mothers with good knowledge of HIV (AOR = 2.4;95%CI: 1.4–4.0) | • A child being found HIV positive (AOR = 0.3; 95%CI: 0.1–0.6) | Sex, residence, place of delivery, maternal HIV status at conception, planned pregnancy, age of the household head, household head educational level, monthly income, size of the household |
| | Ebuy 2020 [62] | • Counselled on feeding options: Yes (AOR = 2.01; 95%CI: 1.11–3.65), No (ref) | | | • ANC follow-up: Yes (AOR = 2.77; 95%CI: 1.17–6.55), no (ref) | Place of residence, mother WHO clinical stage, current CD4 count of mothers, mothers' adherence status, provision of postpartum family planning, infants birth weight, infant ARV prophylaxis (NVP syrup) given |
| | Gaitho 2021 [76] | | | | For late HIV diagnosis<br>• Mothers on ART at the time of HIV PCR test: No (AOR = 1.27; 95% CI: 1.18–1.37), yes (ref)<br>• Received infants ARV prophylaxis: no (AOR = 1.43; 95%CI: 1.27–1.61), yes (ref) | Sex, birth weight, mode of delivery, place of delivery |
| | Hampanda 2017 [89] | | • Intimate partner violence (AOR = 0.41; 95%CI: 0.21–0.79), | • Disclosed HIV status to male partner (AOR = 13.73; 95%CI: 3.59–52.49) | • Maternal ART adherence (AOR = 2.28; 95%CI: 1.15–4.55), | Maternal age, parity, completed secondary education, standardized wealth Index, diagnosed with HIV during most recent pregnancy, male partner HIV status: |
| | Kebede 2014[45] | | | | • Mothers on HAART during pregnancy (AOR = 3.4; 95%CI: 1.5–7.3)<br>• Received ARV prophylaxis during pregnancy (AOR = 3.7; 95%CI: 1.5–8.7)<br>• Place of birth: Government health facility (AOR = 2.9; 95%CI: 1.6–5.5), Home (ref) | Prenatal care (ANC), the time mother gets HIV diagnosed, mother prophylaxis at labour, infant prophylaxis at birth |

AOR- Adjusted Odds Ratio, RR-Relative Risk,

* The factors included only significant factors adjusted for confounders.

with EID. However, having an HIV-positive child resulted in lower odds of EID service uptake [92].

**Qualitative synthesis.** Qualitative studies and qualitative aspects of mixed methods studies were thematically analysed. Themes such as health care factors, access to services, partner-related factors, acceptance and disclosure of HIV status, stigma and misconception, knowledge and couples' differences in HIV status, disease progression, fear related to ART and good health were generated [Table 3].

**Health care factors**: Most women and providers believed that health system factors were the most common challenges across PMTCT cascades. This review illustrated how lack of privacy [94,96–98], shortage of staff and HIV test kits [98], negative attitude/behaviour of health workers [39,84], lack of age-specific service [39], long waiting times [39] and lack of supervision [97] were often responsible for low PMTCT services uptake.

Lack of privacy and counselling were crucial factors, particularly during maternal HIV testing and ART initiation [94,96,98], Furthermore, the shortage of healthcare workers (HCWs) in comparison to the number of clients and the irregular availability of HIV test kits were both reported as a challenge [39]. Providers also commonly reported that lack of follow-up supervision for nurses working in the ART clinics, antenatal clinics (ANCs) and maternity wards, as well as for laboratory and health surveillance assistants who provide EID and treatment services as a challenge [97]. Likewise, the absence of a reliable transport system for dried blood spot (DBS) [97] to perform DNA PCR tests for EID at central reference laboratories was also identified as a challenge. It requires specimen transport over long distances from health centres to the central labs where samples are analysed. The long waiting time between DBS sample collection and the processing of results meant there were repeated facility visits by women and associated unnecessary transport costs. This created frustration and worry in women about the health of their children [97].

A range of healthcare-related motivators was described such as free treatment/services, peer support, client motivation emotional support in health facilities, implementation of SMS and a good referral system.

**Access to facilities/services**: Long distances to the health facility [39,98] and financial constraints [39,80] were reported by women as a barrier to accessing maternal HIV testing and ART initiation. However, a study conducted in Uganda reported that having access to free treatment and services was a strong motivator for the uptake of PMTCT services [39].

**Partner-related factors**: Domestic violence by a partner, lack of partner support, and blame when using ART [39,80,84,85,88,98,101] were reported by women. Women feared domestic violence and were not happy to disclose their HIV status and initiate treatment [85,98]. Likewise, women were usually frustrated with the continuity of their relationship if their partners knew they were HIV positive and on ART [85,88,98,101].

**Stigma and misconceptions**: Five qualitative studies with a focus on HIV testing and ART uptake among pregnant women suggested that HIV-related stigma and misconceptions prevented them from service use [39,80,85,86,98]. Women experience stigma when they are seen testing for HIV and taking ARV medication. In addition, the embodied misconception of HIV such as traditional health beliefs and practices [80,86] as well as religious views [80,85,98] towards HIV played a negative role in the uptake of HIV testing and ART.

**Knowledge and couples' difference in HIV status**: women lacking comprehensive knowledge about HIV and the benefits of ART were identified as a barrier [39,86,96]. Moreover, in couples where women are infected with HIV but their male partners are HIV-negative, the women often experience emotional and psychological distress [88]. This emotional and psychological distress delayed their acceptance and initiation of ART. Therefore, when women

**Table 3. A summary qualitative synthesis of factors to PMTCT services uptake in East Africa.**

| Findings | First author, publication year, and reference number | | | | | | | | | | | | Total |
|---|---|---|---|---|---|---|---|---|---|---|---|---|---|
| | Konje 2018 [94] | Oshosen 2021[96] | Cataldo 2017[98] | Mustapha 2018[39] | Buleza Lamucene 2022[80] | Mukose 2021[86] | Buregyeya 2017[85] | Kanguya 2022[88] | Kim 2016 [101] | Chadambuka 2018[100] | Bergmann 2017[84] | Bobrow 2016[97] | |
| **HIV testing** | | | | | | | | | | | | | |
| **Healthcare-related factors** | | | | | | | | | | | | | |
| Lack of privacy and inadequate counselling | ✓ | ✓ | | | | | | | | | | | 2 |
| Shortage of Staff and HIV test kits | | | ✓ | | | | | | | | | | 1 |
| The negative attitude of health workers | | | | ✓ | | | | | | | | | 1 |
| Long waiting time at the clinic | | | | ✓ | | | | | | | | | 1 |
| Lack of age-specific service for adolescent | | | | ✓ | | | | | | | | | 1 |
| Emotional Support by nurses* | | ✓ | | | | | | | | | | | 1 |
| **Access to facilities/ services** | | | | | | | | | | | | | |
| Long distance to the health facility | | | | ✓ | | | | | | | | | 1 |
| **Acceptance of HIV status** | | | | | | | | | | | | | |
| Fear or denial to accept that they are HIV-positive | ✓ | ✓ | | ✓ | | | | | | | | | 3 |
| **Stigma and misconceptions** | | | | | | | | | | | | | |
| Belief witchcraft is the origin of the disease | | | | | ✓ | | | | | | | | 1 |
| Stigma and discrimination | | | | ✓ | | | | | | | | | 1 |
| **Disease progression** | | | | | | | | | | | | | |
| Being asymptomatic state | | | | ✓ | ✓ | | | | | | | | 2 |
| **Maternal ART uptake** | | | | | | | | | | | | | |
| **Healthcare-related factors** | | | | | | | | | | | | | |

(*Continued*)

**Table 3.** (Continued)

| Findings | First author, publication year, and reference number | | | | | | | | | | | | Total |
|---|---|---|---|---|---|---|---|---|---|---|---|---|---|
| | Konje 2018 [94] | Oshosen 2021[96] | Cataldo 2017[98] | Mustapha 2018[39] | Buleza Lamucene 2022[80] | Mukose 2021[86] | Buregyeya 2017[85] | Kanguya 2022[88] | Kim 2016 [101] | Chadambuka 2018[100] | Bergmann 2017[84] | Bobrow 2016[97] | |
| Lack of privacy/ confidentiality in ART services | | | ✓ | | | | | | | | | | 1 |
| Motivation by a health worker to start treatment* | | | | | | | ✓ | | | | | | 1 |
| Time to ART initiation (Same day ART initiation) * | | | ✓ | | | | | | ✓ | | | | 2 |
| Support and counselling by peer educators * | | | | ✓ | ✓ | | | | | | | | 2 |
| Support and counselling/ health education by health professionals* | | | | ✓ | | ✓ | ✓ | | | ✓ | | | 4 |
| **Access to facilities/ services** | | | | | | | | | | | | | |
| Long distance to a health facility | | | ✓ | | | | | | | | | | 1 |
| Financial constraints | | | | ✓ | ✓ | | | | | | | | 2 |
| Access to free treatment and services* | | | | ✓ | | | | | | | | | 1 |
| **Partners related factors** | | | | | | | | | | | | | |
| Domestic violence and abandonment | | | ✓ | | | | ✓ | | | | | | 2 |
| Lack of partners' support | | | | ✓ | ✓ | | | ✓ | | | | | 3 |
| Partner blame or divorce. | | | ✓ | | | | ✓ | ✓ | ✓ | | | | 4 |
| **Acceptance and disclosure of HIV status** | | | | | | | | | | | | | |
| Doubt of HIV positive results. | | | | | | ✓ | | | | | | | 1 |
| HIV status discloser to their partners | | | | | | | | | | ✓ | | | 1 |
| **Stigma and misconceptions** | | | | | | | | | | | | | |
| Religious belief | | | ✓ | | ✓ | | ✓ | | | | | | 3 |

(*Continued*)

**Table 3.** (Continued)

| Findings | Konje 2018 [94] | Oshosen 2021[96] | Cataldo 2017[98] | Mustapha 2018[39] | Buleza Lamucene 2022[80] | Mukose 2021[86] | Buregyeya 2017[85] | Kanguya 2022[88] | Kim 2016 [101] | Chadambuka 2018[100] | Bergmann 2017[84] | Bobrow 2016[97] | Total |
|---|---|---|---|---|---|---|---|---|---|---|---|---|---|
| Preference for local herbs. | | | | | | ✓ | | | | | | | 1 |
| HIV-related stigma and discrimination | | | ✓ | | ✓ | | ✓ | | | | | | 3 |
| **Knowledge and couples' differences in HIV status** | | | | | | | | | | | | | |
| Lack of knowledge about the purpose of ART | | ✓ | | ✓ | | ✓ | | | | | | | 3 |
| Knowing the benefits of ART* | | | | | | ✓ | ✓ | | | ✓ | | | 3 |
| Discordant couples | | | | | | | | ✓ | | | | | 1 |
| **Disease progression** | | | | | | | | | | | | | |
| Asymptomatic state/feeling healthy | | | | | ✓ | | | | ✓ | | | | 2 |
| **Fear related to ART** | | | | | | | | | | | | | |
| Fear to take ART for lifelong | | | | ✓ | | ✓ | ✓ | | ✓ | | | | 4 |
| Fear of drug side effects | | | | ✓ | ✓ | ✓ | | | | | | | 3 |
| Fear of the big size of the tablets | | | | | | | ✓ | | | | | | 1 |
| **Good Health** | | | | | | | | | | | | | |
| Mothers desire to have an HIV-negative baby* | | | | ✓ | | ✓ | ✓ | ✓ | | ✓ | | | 5 |
| Mother desires to remain healthy* | | | | | | ✓ | ✓ | | | ✓ | | | 3 |
| **Infant ARV prophylaxis** | | | | | | | | | | | | | |
| **Healthcare-related factors** | | | | | | | | | | | | | |
| The bad behaviours of healthcare workers | | | | | | | | | | | ✓ | | 1 |
| Financial and educational benefits that support groups provided* | | | | | | | | | | | ✓ | | 1 |

*(Continued)*

**Table 3.** (Continued)

| Findings | First author, publication year, and reference number | | | | | | | | | | | | Total |
|---|---|---|---|---|---|---|---|---|---|---|---|---|---|
| | Konje 2018 [94] | Oshosen 2021[96] | Cataldo 2017[98] | Mustapha 2018[39] | Buleza Lamucene 2022[80] | Mukose 2021[86] | Buregyeya 2017[85] | Kanguya 2022[88] | Kim 2016 [101] | Chadambuka 2018[100] | Bergmann 2017[84] | Bobrow 2016[97] | |
| **Partners related factors** | | | | | | | | | | | | | |
| Lack of partner support | | | | | | | | | | | ✓ | | 1 |
| **Early infant diagnosis** | | | | | | | | | | | | | |
| **Healthcare-related factors** | | | | | | | | | | | | | |
| Lack of follow-up supervision | | | | | | | | | | | | ✓ | 1 |
| Lack of space at health centers | | | | | | | | | | | | ✓ | 1 |
| Good referrals and coordinating system* | | | | | | | | | | | | ✓ | 1 |
| Community health education about EIDT* | | | | | | | | | | | | ✓ | 1 |
| New PMTCT guidelines (enrolling HEIs into the HIV care system immediately after delivery) * | | | | | | | | | | | | ✓ | 1 |
| SMS system to receive HIV results* | | | | | | | | | | | | ✓ | 1 |
| Follow-up system at the health center* | | | | | | | | | | | | ✓ | 1 |
| No reliable transport system, long distance to facility, lack of fuel for specimen transport, and cost of transport | | | | | | | | | | | | ✓ | 1 |
| **Acceptance and disclosure of HIV status** | | | | | | | | | | | | | |
| Refusal of HIV status | | | | | | | | | | | | ✓ | 1 |
| Lack of HIV status disclosure to their partners | | | | | | | | | | | | ✓ | 1 |
| *Women/child health records* | | | | | | | | | | | | | |

(Continued)

**Table 3.** (Continued)

| Findings | First author, publication year, and reference number | | | | | | | | | | | | Total |
|---|---|---|---|---|---|---|---|---|---|---|---|---|---|
| | Konje 2018 [94] | Oshosen 2021[96] | Cataldo 2017[98] | Mustapha 2018[39] | Buleza Lamucene 2022[80] | Mukose 2021[86] | Buregyeya 2017[85] | Kanguya 2022[88] | Kim 2016 [101] | Chadambuka 2018[100] | Bergmann 2017[84] | Bobrow 2016[97] | |
| Do not bring relevant health information, including HIV status, to the health facilities | | | | | | | | | | | | ✓ | 1 |

*facilitating/motivating factors.

are in this situation, counselling support is needed to encourage them to disclose their HIV-positive status to their HIV-uninfected partners and utilize ART services.

**Disease progression**: women who acquire HIV does not show sign and symptoms at the early stage of infection and are usually asymptomatic or clinically healthy at the time of diagnosis [80,101]. Hence, when women show no symptoms of AIDS, they perceive themselves as healthy and may not accept their HIV-positive status and start ART promptly to prevent vertical transmission.

**Fear related to ART**: Our review also observed that women do not initiate ART because of the fear they have towards ART drugs; such as fear of potential ART side effects (3 studies) [39,80,86], fear of lifelong commitment to taking ART (4 studies) [39,85,86,101], and fear of its size (1 studies) [85]. The studies conducted in Uganda, [39,85,86] and Malawi [101] reported that most mothers were fearful of taking ART daily for their entire life which deters them from starting the treatment. Furthermore, women perceived that if they were not adherent after treatment had started, they would die earlier, so they did not want to start treatment at all since they were not sure that they would be adherent [85].

**Good health**: Women's desire to be healthy [85,86,100] and an interest to protect their unborn children from acquiring HIV infection [39,85,86,88,100] were identified as strong motivators to initiate ART. These studies indicated that the interest of women to be healthy and to have an HIV-free child inspired them to initiate ART uptake because mothers who are highly concerned about their unborn child are more likely to utilize the services hoping that the child would be found negative in the end.

## Discussion

This systematic review analysed the uptake and determinants of PMTCT of HIV services in East Africa. Accordingly, the overall pooled proportion was 82.69% for maternal HIV testing; 88.33% for maternal ART uptake; 84.98% for infant ARV prophylaxis; and 68.77% for EID in East Africa. The finding was found to be promising, however, much work remains to be done to achieve the UNAID's target by 2030 [8]. Besides, in comparison with the other PMTCT cascades, lower EID uptake was observed. This could be due to its complexity as it requires molecular techniques to detect viral nucleic acid rather than serological methods [42]. In resource-constrained settings, performing DBS sample analysis at central reference laboratories is challenging, as it necessitates skilled personnel and complex lab equipment. Moreover, the transportation of specimens over long distances adds to the difficulty, leading to extended turnaround times for test results to be at health facilities [42,105].

The findings of the sub-group analysis showed variation in the level of uptake from country to country. This could be due to differences in socioeconomic characteristics, considerable variation in the timing of PMTCT policies adoption and the extent to which policies are implemented within health facilities [49] along with availability and quality of the service between countries [106–108]. Besides, in comparison with community-based studies higher uptake was observed in facility-based studies as facility settings studies are expected to involve women who have access to health services with ongoing PMTCT services and awareness programs.

The quantitative component of the review pointed to a wide range of community and health care factors (such as place of residence/geographical location, stigma, quality of PMTCT services), and predisposing factors (such as maternal age, child age, education status, religion, marital status, parity, ethnicity, and perceived benefits/ self-efficacy of PMTCT). Along with enabling factors (such as wealth, distance to a health facility, employment status and income) and need factors and prior health services use (such as ANC follow-up, ART adherence, child HIV status, desire to have a child, and STIs symptoms). Some of the factors identified through quantitative studies were consistent with the synthesized findings obtained from the qualitative studies. These included lack of access to facilities/services, disclosure of HIV status, lack of age-specific service, stigma, and lack of knowledge about PMTCT services. The qualitative findings also pointed to additional factors not identified by the quantitative studies. These included the shortage of resources, lack of follow-up supervision, lack of privacy and confidentiality, fear related to ART, being asymptomatic and being a discordant couple.

Amongst the community factors, residence was associated with maternal HIV testing [25,57,58,63,66,103]. However, its effect was inconsistent in that Alemu et al. [25], Astawesegn et al. [57], Yaya et al. [103] and Ejigu et al. [63] found urban residents were more likely to be HIV tested, whilst Abtew et al. [58] and Gebresillassie et al. [66] revealed rural residents were more likely to be HIV tested. The finding of increased maternal HIV testing among urban mothers may be due to the availability of more healthcare centres, and a shorter distance to these centres in urban areas in comparison with rural areas [109,110]. Whereas, the opposite finding may reflect the belief of women from rural areas that their doctor/nurse will react negatively to their refusal-thus they do not opt out of HIV testing unless adequately informed about the opt-out policy by healthcare professionals [111].

This review also identified stigma and discrimination as the most prominent barrier deterring women from HIV test uptake. Social stigmatization may result in difficulty to attend regular clinic visits and further reduces women's opportunities for a social support system that facilitates disclosure of their HIV status and the subsequent decision to take PMTCT services. Therefore, efforts should be directed at community education about HIV to change communities' views about HIV from a fearsome death sentence to a manageable chronic condition [112].

Likewise, among the predisposing factors, never-married, widowed, or divorced women were less likely to take both maternal and infant HIV test services in comparison to their married counterparts [63,75,92]. This may be linked to the lack of psychosocial and financial support that husbands may provide [113]. Moreover, being widowed/divorced/separated women is socially unacceptable in most developing communities [113,114] and therefore women may fear discrimination and feel ashamed of receiving PMTCT services [113].

There is an interplay between higher educational attainment for women and their husbands/partners and higher SES (employment, income, and wealth index status) in influencing maternal HIV test uptake [25,57,59,63,66,95,103]. This is because, the more educated women are, the greater their level of employment and financial independence and the better informed they are about the importance of PMTCT services [115]. A similar effect of SES on maternal and child health service use has been documented in other studies [116,117]. The more a

community is advantaged socioeconomically, the higher the likelihood of women in that community utilizing health services [118,119]. This finding highlights the need to promote universal primary education and improve women's SES.

The present review demonstrated that knowledge and awareness about HIV and PMTCT were significantly associated with maternal HIV testing [25,57,63–65,75] and EID [92]. Individuals' knowledge and awareness of health issues influenced their: perceived need, perceived services benefit and recognition of the healthcare service centres which provided health services. Therefore, repeated PMTCT messages through media to improve women's knowledge and motivate them to use PMTCT services for themselves and their children, especially in developing countries is recommended [120]. Whereas, long travelling distances to access health facilities providing PMTCT services were associated with lower odds of PMTCT services utilization [57,59,92], because, during pregnancy, walking or travelling long distances is very difficult and may discourage women from using the services in addition to travel-related costs.

Regarding the association between need factors and prior health services use with PMTCT service uptake, there was clear evidence for a positive association between these factors. In line with studies conducted in Ghana [121], ANC follow-up and facility delivery were found to be important factors associated with PMTCT service uptake [62,64,65,84,95]. ANC follow-up and facility delivery create an opportunity for early screening and enrolment of mothers and their newborns into the PMTCT service [19]. Furthermore, studies demonstrated that a child who has used ARV prophylaxis [45,76] or was born from a mother who took ART/HAART [76] had higher odds of testing for HIV at six weeks. Women on ART and infants on ARV prophylaxis are more likely to access EID services because of good awareness of their HIV status, more frequent medical visits, and increased opportunities for providers to identify HEIs. Therefore, attendance and integration of maternal and child health (MCH) services (e.g., ANC) with the PMTCT program is an important strategy to eliminate HIV infection among children [75].

Concerning findings from the qualitative component, healthcare factors (such as shortage of resources, lack of privacy and confidentiality, and lack of follow-up supervision) were identified as a barrier. For example, staff shortages may mean that healthcare providers are unable to provide services to all women in need, which can be experienced directly by women as neglected. Poor infrastructure may also create stressful working environments, which may predispose healthcare providers to behave poorly towards women [94,98]. Similarly, the absence of follow-up supervision was also identified by HCWs as a challenge in EID service delivery. If HCWs are not supervised and trained, they may not have the necessary skills or knowledge of current treatment protocols or referral procedures for providing EID. Hence, ensuring closer supervision of health workers and provision of the needed work inputs and training could improve the utilization of PMTCT services. Furthermore, consistent with a previous systematic review [122], fear related to ART, and being asymptomatic have been also associated with lower ART initiation. Several factors were identified as facilitators or motivators including same-day ART initiation, support/counselling/motivation by health workers or peer educators, free treatment/services, mother's wish to be healthy or to have an HIV-free child, application of SMS system, good referrals and coordinating systems. It is important to note that even though same-day ART initiation is an important factor to increase ART initiation it has been reported to result in poor adherence and treatment discontinuation [122] because women may not have adequate time to think and make an informed decision before initiating ART.

This systematic review has several strengths. Firstly, we used a comprehensive approach to capture all possible articles on this review question. Secondly, we have included articles conducted with quantitative, qualitative, and mixed-methods study designs, without restricting

any study design/method. The use of both quant and qualitative evidence is the best approach to inform policies [55]. Thirdly, we explored whether there were differences in the level of uptake and factors across key PMTCT cascades. As a limitation, firstly, we used a narrative synthesis of factors associated with PMTCT cascades due to wide variations in the measurement of variables among studies. Secondly, we did not search for grey literature, we did not consider publications in other languages apart from English.

## Conclusion

In conclusion, the pooled uptake of the PMTCT service cascade was promising in East Africa, but this finding should be interpreted with caution mainly because of the high between-study variability. Ensuring women and their children are enrolled and retained across the PMTCT cascade is recommended. The most identified factors associated with the service uptake were residence, educational status of parents, SES, stigma towards HIV-positive women, marital status, knowledge on PMTCT, intimate partner violence, attitudes/perceived benefits towards PMTCT services, lack of access to PMTCT service and healthcare-related factors like resource scarcity and insufficient follow-up supervision. These factors are modifiable by deliberately focusing on addressing them systematically, both at the policy and service delivery levels. Therefore, it is advisable to promote women's education and economic empowerment while reducing stigma through active community involvement. Additionally, strengthening staff supervision and improving access to PMTCT services, integrating with maternal and child health care are recommended. Engaging community health workers and expert mothers can also help to share the workload of healthcare providers because of human resource shortages.

## Supporting information

**S1 Fig. Publication bias.**
(TIF)

**S1 Table. PRISMA 2009 checklist.**
(DOCX)

**S2 Table. Search strategy.**
(DOC)

**S3 Table. Methodological quality assessment.**
(DOCX)

**S4 Table. Subgroup analysis.**
(DOCX)

## Acknowledgments

The authors would like to thank Mr Tesfaye Yitna from Wolaita Sodo University, School of Nursing for his assistance in the study selection.

## Author Contributions

**Conceptualization:** Feleke Hailemichael Astawesegn, Haider Mannan, Virginia Stulz, Elizabeth Conroy.

**Data curation:** Feleke Hailemichael Astawesegn.

**Formal analysis:** Feleke Hailemichael Astawesegn, Virginia Stulz.

**Methodology:** Feleke Hailemichael Astawesegn, Haider Mannan, Virginia Stulz, Elizabeth Conroy.

**Project administration:** Feleke Hailemichael Astawesegn, Elizabeth Conroy.

**Software:** Feleke Hailemichael Astawesegn.

**Supervision:** Haider Mannan, Virginia Stulz, Elizabeth Conroy.

**Writing – original draft:** Feleke Hailemichael Astawesegn.

**Writing – review & editing:** Feleke Hailemichael Astawesegn, Haider Mannan, Virginia Stulz, Elizabeth Conroy.

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
