## [Decision Letter · Decision Letter 0]

25 Jul 2023

PONE-D-23-04697Understanding the uptake and determinants of prevention of mother-to-child transmission of HIV services in East Africa: Mixed methods systematic review and meta-analysisPLOS ONE

Dear Dr. Astawesegn,

Thank you for submitting your manuscript to PLOS ONE. After careful consideration, we feel that it has merit but does not fully meet PLOS ONE’s publication criteria as it currently stands. Therefore, we invite you to submit a revised version of the manuscript that addresses the points raised during the review process.

We look forward to receiving your revised manuscript.

Kind regards,

Felix Apiribu, Ph. D., MPhil

Academic Editor

PLOS ONE

Journal Requirements:

Additional Editor Comments (if provided):

The author should do the revisions and submit for final decision.

Reviewers' comments:

Reviewer's Responses to Questions

**Comments to the Author**

1. Is the manuscript technically sound, and do the data support the conclusions?

Reviewer #1: Yes

Reviewer #2: Yes

Reviewer #3: Yes

2. Has the statistical analysis been performed appropriately and rigorously? 

Reviewer #1: Yes

Reviewer #2: Yes

Reviewer #3: Yes

3. Have the authors made all data underlying the findings in their manuscript fully available?

Reviewer #1: Yes

Reviewer #2: Yes

Reviewer #3: Yes

4. Is the manuscript presented in an intelligible fashion and written in standard English?

Reviewer #1: Yes

Reviewer #2: Yes

Reviewer #3: Yes

5. Review Comments to the Author

Reviewer #1: This paper is well written and addresses an important question that must be addressed to reach HIV epidemic control goals for countries in East Africa.

The paper would benefit from a time-stratified analysis, differentiating between HIV service delivery approaches before and after the World Health Organization's "Treat All" guidelines in 2016. Access to HIV testing and treatment dramatically advanced with this guidance, with many women receiving HIV diagnosis and treatment prior to pregnancy which is known to be superior to diagnosis and initiation of treatment during pregnancy from a PMTCT perspective. As presented, the analysis compiles findings from studies that were conducted in dramatically different HIV service delivery contexts and may not present the most relevant findings for programs.

Reviewer #2: The manuscript is well written with minor corrections inserted in the attached. I would be willing to review the revised manuscript,although I don't think it is very necessary given the fact the authors have done a very commendable research work.

Reviewer #3: I appreciate the authors contributions for this well written manuscript of public health importance.

Authors should provide the confidence intervals of the Adjusted Odd Ratios (AOR).

Addressing the strengths and the limitations of the manuscript is commendable.

Conclusively, manuscript respects scientific norms for publication.

6. PLOS authors have the option to publish the peer review history of their article (what does this mean?). If published, this will include your full peer review and any attached files.

Reviewer #1: No

Reviewer #2: No

Reviewer #3: **Yes: **Professor Halle-Ekane Gregory (MD;FWACS)

---

## [Author Response · Author response to Decision Letter 0]

6 Aug 2023

Date: 06/08/2023

Title: Understanding the uptake and determinants of prevention of mother-to-child transmission of HIV services in East Africa: Mixed methods systematic review and meta-analysis 

Dear Editor and Reviewers: We really thank you for your valuable comments/suggestions for the betterment of our manuscript. Your concerns and suggestions are addressed in the revised manuscript and below we have tried to clarify some of the points raised by the reviewers. 

Reviewer # 1

1. The paper would benefit from a time-stratified analysis, differentiating between HIV service delivery approaches before and after the World Health Organization's "Treat All" guidelines in 2016. Access to HIV testing and treatment dramatically advanced with this guidance, with many women receiving HIV diagnosis and treatment before pregnancy which is known to be superior to diagnosis and initiation of treatment during pregnancy from a PMTCT perspective. As presented, the analysis compiles findings from studies that were conducted in dramatically different HIV service delivery contexts and may not present the most relevant findings for programs.

Author response: As you mentioned, there has been a significant shift in the HIV guidelines, such as moving away from considering WHO stage or CD4 count levels to adopting a test and treat approach for the general population in 2016`. However, when it comes to PMTCT, the approach has evolved differently due to its specific goal of preventing vertical transmission.

Over the last two decades, WHO has developed various PMTCT guidelines/approaches, such as option A, option B and option B+. Option B+ involves a test and treat approach for HIV-positive pregnant women only with the primary objective of preventing vertical transmission from mother to child and has been in effect since 2012 (Malawi was the first country to implement this in East Africa). Therefore, given the focus of this review on understanding the current status of preventing mother-to-child HIV transmission, articles published after the implementation of Option B+ (since 2012) were included in this review to gather relevant information.

Reviewer # 2

2. To my understanding Ethiopia, Mozambique, Zimbabwe, etc are not part of East Africa. I suggest that authors should review or justify the inclusion of these countries.

Author response: Regarding the country's classification, there are different classifications of East African countries. In this study, we selected the United Nations Statistical Division and World Health Organization classification of East Africa region as mostly health-related reports and studies considered this classification. We have cited the following reference to indicate the list of East African countries in the manuscript method section (see page 5 lines 128-129). 

United Nations Department of Economic and social affairs statistical division. Standard country or area codes for statistical use 2020 [Available from: https://unstats.un.org/unsd/methodology/m49/

United Nation. World Population Review 2020 [Available from: https://worldpopulationreview.com/country-rankings/east-african-countries

Furthermore, recently published studies (including those published by Plos One) using the same countries under East Africa are cited below for the reviewer's consideration: 

1. Tesema GA, Tessema ZT. Pooled prevalence and associated factors of health facility delivery in East Africa: mixed-effect logistic regression analysis. PLoS One. 2021 Apr 23;16(4):e0250447.

2. Raru TB, Ayana GM, Zakaria HF, Merga BT. Association of higher educational attainment on antenatal care utilization among pregnant women in East Africa using Demographic and Health Surveys (DHS) from 2010 to 2018: a multilevel analysis. International journal of women's health. 2022 Feb 1:67-77.

3. From the review findings, the authors can suggest specific strategies/implemented target interventions.

Author response: Thank you so much. We have included specific strategies/implemented target interventions as part of the discussion and conclusion ( see page 30 lines 537-541)

4. Was the protocol of this systematic review registered? Authors should provide information.

Author response: We were unable to register the protocol because of the time limitations we encountered. During that time, we encountered difficulty in registering a protocol on platforms such as PROSPERO, due to the vast amount of reviews related to COVID-19 that had overloaded them.

5. Authors should clarify the meta-analyses mentioned here in relation to the following statement: "...it was not feasible to conduct a meta-analysis to assess the effect of each factor." 

Author response: Thank you so much. To clarify this, a meta-analysis was done to estimate PMTCT services uptake in percentage, as we have indicated under the method section (page 7 lines 198-199). The result of the meta-analysis for the service uptake has been presented on page 14. Whereas, for the determinants or factors associated with PMTCT service uptake, we did not conduct a meta-analysis because of wide variations in the measurement and classification of the variables between studies (see pages 7 line number 191-194). The result of narrative synthesis for the factor has been presented on pages 15-26

6. General comment: In comparison with previous systematic reviews on the same topic, what are new findings that this review has found different from those that have filled the existing knowledge gap?

Author response: Unlike other reviews, this review holistically addressed the uptake and determinants of key PMTCT services cascade such as (i) maternal HIV testing among pregnant/postpartum women, (ii) ART initiation among HIV-infected pregnant/postpartum women, (iii) initiation of ARV prophylaxis for HIV exposed infants (HEIs), and (iv) early infant diagnosis (EID)/HIV test at six weeks of age/ in East Africa. It enables readers to compare uptake and determinants across the cascade. Most previous systematic reviews included only quantitative studies, whereas this review included studies that employed mixed methods (quantitative and qualitative). after the implementation of option B+. Most previous studies done after the implementation of option b+ were focused on determinants of ART adherence, however, this review focuses on determinants of PMTCT cascade uptake 

7. Can authors justify, the reasons for not searching for grey literature, and if it has affected their review findings?

Author response: To focus on peer-reviewed and published articles. 

8. What about limited number of HCWs??

Author response: We have provided suggestions to improve the number of HCWs under the conclusion section ( page 30, lines 540-541)

Reviewer #3

9. Authors should provide the confidence intervals of the Adjusted Odd Ratios (AOR). 

Author response: Thank you, we have addressed this omission in the revised manuscript table 2 (pages 17-21).

We appreciate the editor and reviewers for the comments and time taken to read our manuscript. And hope that we have addressed the reviewers’ comments and associated changes in the manuscript to your satisfaction.

Thank you. 

Feleke Hailemichael

---

## [Decision Letter · Decision Letter 1]

2 Feb 2024

PONE-D-23-04697R1Understanding the uptake and determinants of prevention of mother-to-child transmission of HIV services in East Africa: Mixed methods systematic review and meta-analysisPLOS ONE

Dear Dr. Astawesegn,

Thank you for submitting your manuscript to PLOS ONE. After careful consideration, we feel that it has merit but does not fully meet PLOS ONE’s publication criteria as it currently stands. Therefore, we invite you to submit a revised version of the manuscript that addresses the points raised during the review process.

Please attend to all the outstanding comments raised by the reviewers 

We look forward to receiving your revised manuscript.

Kind regards,

Hamufare Dumisani Dumisani Mugauri, Ph.D. Public Health

Academic Editor

PLOS ONE

Journal Requirements:

Reviewers' comments:

Reviewer's Responses to Questions

**Comments to the Author**

1. If the authors have adequately addressed your comments raised in a previous round of review and you feel that this manuscript is now acceptable for publication, you may indicate that here to bypass the “Comments to the Author” section, enter your conflict of interest statement in the “Confidential to Editor” section, and submit your "Accept" recommendation.

Reviewer #3: All comments have been addressed

Reviewer #4: All comments have been addressed

2. Is the manuscript technically sound, and do the data support the conclusions?

Reviewer #3: Yes

Reviewer #4: Yes

3. Has the statistical analysis been performed appropriately and rigorously? 

Reviewer #3: Yes

Reviewer #4: Yes

4. Have the authors made all data underlying the findings in their manuscript fully available?

Reviewer #3: Yes

Reviewer #4: Yes

5. Is the manuscript presented in an intelligible fashion and written in standard English?

Reviewer #3: Yes

Reviewer #4: Yes

6. Review Comments to the Author

Reviewer #3: I have no additional comments for the authors. Furthermore, the comments of the other reviewers, responses, and the modifications made by the authors will improve on the quality of the manuscript.

Thank you

Reviewer #4: This paper is a systematic review which can provide insights on uptake of PMTCT in the East Africa; indeed the findings may be useful to move forward elimination of MTCT,assuming validity of the conclusion and recommendations. The key innovation is the mixed method both quantitave and qualitative for this systematic review.

RESULTS

The high variability of the studies included in this meta-analysis makes it difficult to summarize the results and recommendations.Undoubtedly the work was exhaustive, however: despite the detailed description of the objective of the studies considered for the quantitative pole ( column 2 table 1),

1/the main measurement(s) for each of the studies does not appear in Table 1. Several selected studies focus on on antenatal screening, antenatal monitoring and early diagnosis; proportionally fewer studies target antiretroviral treatment during pregnancy or afterward.

2/Classifying the studies in Table 1 according to the target of the PMTCT cascade could be recommended? if possible

DISCUSSION AND STUDY LIMITATIONS

Within the limits of the study, in addition to those cited, the authors could mention the failure to take into account in their cascade the time of final diagnosis at 18 months of children exposed to HIV;Because, it is the central indicator for the elimination of PMTCT as well as ithey could mention not having taken into account the questions of adherence to antiretroviral treatment of pregnant/lactating women and the retention of mother babies couples in follow-up up to 18 -24 months ( the end of PMTCT CASCADE)

In the conclusion and recommendations line 540 541, while mentioning expert mothers, authors should reinforce their references with the experience of mothers 2 mothers. (as this is not coming from the findings of their review)

REFERENCES

have problem with reference 122 which describes PMTCT in a specific context of key population, here sex workers....which are quite different in profile than general population of pregnant women ....

7. PLOS authors have the option to publish the peer review history of their article (what does this mean?). If published, this will include your full peer review and any attached files.

Reviewer #3: **Yes: **Professor Halle-ekane Gregory (MD, FWACS)

Reviewer #4: **Yes: **ANNE ESTHER NJOM NLEND

---

## [Author Response · Author response to Decision Letter 1]

24 Feb 2024

Date: 15/02/2024

Title: Understanding the uptake and determinants of prevention of mother-to-child transmission of HIV services in East Africa: Mixed methods systematic review and meta-analysis 

Dear Editor and Reviewers: We really thank you for your valuable comments/suggestions for the improvement of our manuscript. Your concerns and suggestions are addressed in the revised manuscript and below we have tried to clarify some of the points raised by the reviewers. 

Reviewers comment

Reviewers: The high variability of the studies included in this meta-analysis makes it difficult to summarize the results and recommendations. Undoubtedly the work was exhaustive, however: despite the detailed description of the objective of the studies considered for the quantitative pole (column 2 Table 1),

1/the main measurement(s) for each of the studies does not appear in Table 1. Several selected studies focus on antenatal screening, antenatal monitoring, and early diagnosis; proportionally fewer studies target antiretroviral treatment during pregnancy or afterwards. 2/Classifying the studies in Table 1 according to the target of the PMTCT cascade could be recommended. if possible

Authors: Thank you for your comment. I have included targeted PMTCT cascades in Table 1 column 7 with footnote description. 

Reviewers: Within the limits of the study, in addition to those cited, the authors could mention the failure to take into account in their cascade the time of final diagnosis at 18 months of children exposed to HIV; Because it is the central indicator for the elimination of PMTCT as well as they could mention not having taken into account the questions of adherence to antiretroviral treatment of pregnant/lactating women and the retention of mother babies couples in follow-up up to 18 -24 months ( the end of PMTCT CASCADE)

Authors: We appreciate the reviewer's concern. Of course, in places where there is no virological testing, babies typically must wait until they are 18-24 months old to undergo an antigen-antibody test. This is because it's expected that maternal antigen will clear out of the infant's blood by 18-24 months, allowing for accurate antigen tests. However, this review focused on the recommended "early" infant diagnosis, which involves testing HIV-exposed infants within 6 weeks of birth. Early infant diagnosis employs a virological test using DNA-PCR to directly detect HIV. This suggestion is advised to prevent infant deaths, as infants with HIV have a high likelihood of death because of the disease before reaching the age of two. It is advised for countries to establish virological testing using HIV-DNA (PCR) and/or HIV-RNA at least at central or tertiary levels to enable early HIV testing for infants. Therefore, the study, as outlined in the methods section, aims to examine the uptake and determinants of "early infant diagnosis”, involving virological tests before 6 weeks.

Reviewers: In the conclusion and recommendations line 540 541, while mentioning expert mothers, authors should reinforce their references with the experience of mothers 2 mothers. (As this is not coming from the findings of their review). 

Response: As you mentioned, involving expert mothers while dealing with one mother to another was not the finding of the review, we rather recommended involving expert mothers as a solution for the staff shortage we identified in this review. Expert mothers (EMs) are mothers living with HIV who have personal experience with PMTCT programs, which they can draw upon to support other women and their families. Previous studies showed that the involvement of expert mothers in formal health systems in low-resource settings was found to be important for overcoming workload through task sharing. They possess the ability to undertake various tasks within facilities and communities aimed at assisting HIV-positive women and their families. EMs frequently serve as auxiliary staff for services facing staffing shortages. Their responsibilities may include administering routine HIV tests typically conducted by healthcare workers (HCWs), offering pre-and post-HIV test counselling, delivering psychosocial support and guidance on treatment adherence, conducting home visits to assess client well-being, sending reminders for clinic appointments, and tracking patients who have been lost to follow up from PMTCT services.

Reviewers: have a problem with reference 122 which describes PMTCT in a specific context of a key population, here sex workers.... which are quite different in profile than general population of pregnant women ....

Response: The point is appreciated, and the reference has been removed. 

We appreciate the editor and reviewers for the comments and time taken to read our manuscript. And hope that we have addressed the reviewers’ comments and associated changes in the manuscript to your satisfaction.

Thank you. 

Feleke Hailemichael

---

## [Editor Report · Decision Letter 2]

1 Mar 2024

Understanding the uptake and determinants of prevention of mother-to-child transmission of HIV services in East Africa: Mixed methods systematic review and meta-analysis

PONE-D-23-04697R2

Dear Dr. Feleke,

We’re pleased to inform you that your manuscript has been judged scientifically suitable for publication and will be formally accepted for publication once it meets all outstanding technical requirements.

Kind regards,

Hamufare Dumisani Dumisani Mugauri, Ph.D. Public Health

Academic Editor

PLOS ONE